# Partially Adaptive Regularized Multiple Regression Analysis for Estimating Linear Causal Effects

**Hisayoshi Nanmo**[1]

**Manabu Kuroki**[2]

[1]Chugai Pharmaceutical Co., Ltd., Nihonbashi Muromachi, Chuo-ku, Tokyo, Japan
[2]Yokohama National University , Tokiwadai, Hodogaya-ku, Yokohama, Japan

## Abstract

This paper assumes that cause-effect relationships among variables can be described with a linear structural equation model. Then, a situation is considered where a set of observed covariates satisfies the back-door criterion but the ordinary least squares method cannot be applied to estimate linear causal effects because of multicollinearity/high-dimensional data problems. In this situation, we propose a novel regression approach, the "partially adaptive $L_p$-regularized multiple regression analysis" ($PAL_pMA$) method for estimating the total effects. Different from standard regularized regression analysis, $PAL_pMA$ provides a consistent or less-biased estimator of the linear causal effect. $PAL_pMA$ is also applicable to evaluating direct effects through the single-door criterion. Given space constraints, the proofs, some numerical experiments, and an industrial case study on setting up painting conditions of car bodies are provided in the Supplementary Material.

## 1 INTRODUCTION

### 1.1 BACKGROUND

The multicollinearity problem [Frisch, 1934], which occurs when two or more explanatory variables are highly correlated, is an important issue in regression analysis. If multicollinearity exists, because the performance of least squares/maximum likelihood estimators of regression coefficients is inadequate, valid results may not be obtained. The high-dimensional data problem occurs in the framework of regression analysis when the sample size is smaller than the number of explanatory variables. High-dimensional data analysis also suffers from multicollinearity, which causes overfitting and interferes with obtaining admissible solutions for regression coefficients. Recently, due to the development of technological advances that help collect data with a large number of variables to better understand a given phenomenon of interest, multicollinearity/high-dimensional data problems have become serious in many domains. To overcome this difficulty, numerous kinds of variable selection techniques based on regularized regression analysis, for example, the least absolute shrinkage and selection operator (LASSO), elastic net, smoothly clipped absolute deviation (SCAD) and minimax concave penalty (MCP) methods, have been proposed by many statistical and AI researchers and practitioners [Bühlmann and van de Geer, 2011; Efron et al, 2004; Fan and Li, 2001; Hoerl and Kennard, 1970; Kuroki and Matsuura, 2018, 2019, 2020; Tibshirani, 1996; van de Geer et al, 2014; Zhang, 2010; Zou, 2006; Zou and Hastie, 2005].

Currently, the role of regression analysis is not limited to the prediction of a response variable by explanatory variables; it also plays an important role in evaluating the linear causal effects of the treatment variable on the response variable. In particular, the total effect, which is one of the representative linear causal effects and the main interest in this paper, means the changes in the expected response variable by one unit through an external intervention [Pearl, 2009, 2013, 2017]. As has often been noted in the framework of statistical causal inference, to derive the consistent estimator of the total effect, in addition to the treatment variable, confounders must be included as explanatory variables in the regression model. However, there are many confounders that have an effect on both the treatment variable and the response variable and that are highly correlated in reality. This situation leads to the multicollinearity problem, which deteriorates the estimation accuracy of the total effects and formulates an

*Accepted for the 38th Conference on Uncertainty in Artificial Intelligence* (UAI 2022).

unreliable plan that prevents us from conducting appropriate policy decision making. On the other hand, the present countermeasures against the multicollinearity problem are formulated independently of the confounding problem. Thus, although stable results of regression analysis may be derived by these countermeasures from the viewpoint of the prediction, they may yield a highly biased estimate of the linear causal effect.

## 1.2 CONTRIBUTIONS

In this paper, when the cause-effect relationships among variables can be described with a linear structural equation model, we consider a situation where a set of observed covariates satisfies the back-door criterion but the ordinary least squares (OLS) method cannot be applied to estimate the total effects because of the multicollinearity/high-dimensional data problem. In this situation, to evaluate the total effect, we propose a novel regression approach, the "Partially Adaptive $L_p$-regularized Multiple regression Analysis" (PAL$_p$MA) method for $p = 1, 2$. In particular, PAL$_1$MA has the following desirable properties:

(1) In statistical causal inference, it is important not to remove a treatment variable or confounders from the regression model when estimating the total effects. However, even if some covariates are guaranteed to be important confounders from qualitative causal knowledge, standard regularized regression analysis may remove them and the treatment variable from the model, depending on the value of the regularization parameter. In contrast, PAL$_1$MA enables us to include both the treatment variable and such covariates in the regression model, regardless of the value taken by the regularization parameter. In particular, when we know that a set of covariates satisfies the back-door criterion, the solution path with such information can be utilized as the criteria of parameter tuning to estimate the total effects.

(2) Regarding PAL$_p$MA for $p = 1, 2$, we can derive a collapsibility condition, i.e., a sufficient condition that the $L_p$-regularized estimator of the regression coefficient of interest is consistent with the OLS estimator regardless of the value taken by the regularization parameter, and thus leads to the consistent estimator of the total effects under the condition. The collapsibility problem in regression analysis have been discussed by many researchers [Clogg et al, 1992; Geng and Asano, 1993; Guo and Geng, 1995; Wermuth, 1989ab]. However, to the best of our knowledge, there has been much less discussion of collapsibility problem in the context of regularized regression analysis.

(3) Compared to standard regularized regression analysis, PAL$_1$MA can reduce the bias or provide higher coincidence rates for the signs of the OLS estimator, even when the collapsibility conditions are violated. In contrast, in standard regularized regression analysis, the regression coefficients can flip from positive to negative values and from negative to positive values as they shrink toward zero, depending on the value of the regularization parameter. This phenomenon implies that standard regularized regression analysis may provide misleading qualitative results regarding the total effects compared to PAL$_1$MA.

From these properties, PAL$_1$MA contributes to solving the multicollinearity/high-dimensional data problems of evaluating linear causal effects in the context of statistical causal inference. Given space constraints, the proofs, some numerical experiments and an industrial case study on setting up painting conditions of car bodies [Kuroki, 2012] are provided in the Supplementary Material.

## 2 LINEAR STRUCTURAL CAUSAL MODEL

In the context of statistical causal inference, a directed acyclic graph that represents cause-effect relationships is called a causal diagram. A directed graph is a pair $G = (\boldsymbol{V}, \boldsymbol{E})$, where $\boldsymbol{V}$ is a finite set of vertices and the set $\boldsymbol{E}$ of directed arrows is a subset of the set $\boldsymbol{V} \times \boldsymbol{V}$ of ordered pairs of distinct vertices. In this paper, we refer to vertices in the directed acyclic graph and random variables of the linear structural equation model interchangeably. In addition, for the graph theoretic terminology used in this paper, we refer readers to Pearl [2009].

**Definition 1** *(Linear Structural Causal Model) Suppose a directed acyclic graph $G = (\boldsymbol{V}, \boldsymbol{E})$ with set $\boldsymbol{V} = \{V_1, V_2, \cdots, V_m\}$ of variables is given. The graph $G$ is called a causal diagram when each child-parent family in the graph $G$ represents a linear structural equation model*

$$V_i = \mu_{v_i} + \sum_{V_j \in pa(V_i)} \alpha_{v_i v_j} V_j + \epsilon_{v_i}, \ \ i = 1, 2, \ldots, m \tag{1}$$

*as the data generating process, where $pa(V_i)$ denotes a set of parents of $V_i$ in $G$ and random disturbances $\epsilon_{v_1}, \epsilon_{v_2}, \ldots, \epsilon_{v_m}$ are assumed to be independent and identically distributed with mean $0$. In addition, $\mu_{v_i}$ is an intercept, and $\alpha_{v_i v_j}(\neq 0)$ is called a path coefficient or a direct effect of $V_j$ on $V_i$ $(i, j = 1, 2, \ldots, m \,;\, i \neq j)$. Then, equation (1) is called a linear structural causal model (SCM) in this paper.*

To proceed with our discussion, we define some notation. For univariates $X$ and $Y$ and a set of variables $\boldsymbol{Z}$, let $\sigma_{xy \cdot z}$ be the conditional covariance between $X$ and $Y$ given $\boldsymbol{Z} = \boldsymbol{z}$, and let $\sigma_{xx \cdot z}$ be the conditional variance of $X$ given $\boldsymbol{Z} = \boldsymbol{z}$. The regression coefficient of $X$ in the regression model of $Y$ on $X$ and $\boldsymbol{Z}$ is denoted by $\beta_{yx \cdot z} = \sigma_{xy \cdot z} / \sigma_{xx \cdot z}$. For sets of variables $\boldsymbol{X}$, $\boldsymbol{Y}$, and $\boldsymbol{Z}$ ($\boldsymbol{Y}$ can be univariate), let $\Sigma_{xy \cdot z}$ be the conditional cross-covariance matrix between $\boldsymbol{X}$ and $\boldsymbol{Y}$ given $\boldsymbol{Z} = \boldsymbol{z}$, and let $\Sigma_{xx \cdot z}$ be the conditional variance-covariance matrix of $\boldsymbol{X}$ given $\boldsymbol{Z} = \boldsymbol{z}$. In addition, let $B_{yx \cdot z} = \Sigma_{xx \cdot z}^{-1} \Sigma_{xy \cdot z}$ denote the regression coefficient vector of $\boldsymbol{X}$ in the regression model of $Y$ on $\boldsymbol{X}$ and $\boldsymbol{Z}$. The set of variables $\boldsymbol{Z}$ is omitted from these arguments if it is an empty set. Similar notation is used for the remaining statistical parameters. Furthermore, letting $\boldsymbol{X} = \{X_1, X_2, ..., X_q\}$, the $i$-th element of $B_{yx \cdot z}$ is denoted by $\beta_{yx_i \cdot x_{(i)} z}$, where $\boldsymbol{X}_{(i)} = \boldsymbol{X} \backslash \{X_i\}$ ($i = 1, 2, ..., q$). $\boldsymbol{0}_q$ is a $q$-dimensional zero vector. Similar notation is used for other sets of variables.

The main purpose of this paper is to estimate total effects from observed data. The total effect $\tau_{yx}$ of $X$ on $Y$ is defined as the total sum of the products of the path coefficients on the sequence of directed arrows along all the directed paths from $X$ to $Y$. To achieve our aim, we introduce the back-door criterion [Pearl, 2009] as one of the representative identifiability criteria for the total effects. Here, when a linear causal effect can be determined uniquely from the variance/covariance parameters of the observed variables, it is said to be identifiable, that is, it can be estimated consistently.

**Definition 2** *(Back-Door Criterion) Let $\{X, Y\}$ and $\boldsymbol{Z}$ be disjoint subsets of $\boldsymbol{V}$ in a directed acyclic graph $G$. If a set $\boldsymbol{Z}$ of vertices satisfies the following conditions relative to an ordered pair $(X, Y)$, then $\boldsymbol{Z}$ is said to satisfy the back-door criterion relative to $(X, Y)$.*

1. *No vertex in $\boldsymbol{Z}$ is a descendant of $X$, and*

2. *$\boldsymbol{Z}$ d-separates $X$ from $Y$ in the graph obtained by deleting all the directed arrows emerging from $X$ from graph $G$.*

If a set $\boldsymbol{Z}$ of observed variables satisfies the back-door criterion relative to $(X, Y)$ in a causal diagram $G$, then, the total effect $\tau_{yx}$ is identifiable and is given by the formula $\beta_{yx \cdot z}$ [Pearl, 2009]. For other identification conditions of linear causal effects, refer to, for example, Brito [2004], Cai and Kuroki [2008], Chan and Kuroki [2010], Chen [2017], Chen et al [2017], Kuroki and Pearl [2014], Pearl [2009], Stanghellini [2004], Stanghellini and Pakpahan [2015] and Tian [2004, 2007ab].

Here, a covariate is defined as an element of non-descendants of $X$ and $Y$. In addition, covariates in a minimal set of variables that satisfy the back-door criterion are called confounders. Note that such a minimal set is not unique and whether or not a certain covariate is considered a confounder depends on the selected minimal set. Furthermore, a set of covariates satisfying the back-door criterion is also called a sufficient set of confounders; otherwise, it is called an insufficient set of confounders. For details on the SCM, refer to the paper by Pearl [2009]. Finally, the direct effect is also known as one of the representative linear causal effects. However, we are concerned with the evaluation of the total effects because the direct effect can also be discussed in the framework of regression analysis through the "single-door criterion" [Pearl, 2009]. Thus, the total effects are identified with linear causal effects in this paper.

## 3 PAL$_p$MA

### 3.1 SETUP

Let $X$, $Y$, $\boldsymbol{Z}$ and $\boldsymbol{W}$ be a treatment variable (and an explanatory variable), a response variable, an $r$-dimensional vector of explanatory variables ($\boldsymbol{Z}$ can be empty) and a $q$-dimensional vector of explanatory variables ($\boldsymbol{W}$ can be empty), respectively. For a sample size of $n$, consider the linear regression model of $Y$ on $X$, $\boldsymbol{Z}$ and $\boldsymbol{W}$

$$\boldsymbol{y} = \boldsymbol{x} \beta_{yx \cdot zw} + \boldsymbol{z} B_{yz \cdot xw} + \boldsymbol{w} B_{yw \cdot xz} + \boldsymbol{\epsilon}_{y \cdot xzw}, \quad (2)$$

where $\boldsymbol{x}$ and $\boldsymbol{y}$ represent $n$-dimensional observation vectors of $X$ and $Y$, respectively. In addition, $\boldsymbol{z}$ and $\boldsymbol{w}$ are an $n \times r$ observation matrix of $\boldsymbol{Z}$ and an $n \times q$ observation matrix of $\boldsymbol{W}$, respectively. Furthermore, $\beta_{yx \cdot zw}$, $B_{yz \cdot xw}$ and $B_{yw \cdot xz}$ are the regression coefficient of $X$, the regression vector of $\boldsymbol{Z}$ and the regression vector of $\boldsymbol{W}$ in equation (2), respectively. $\boldsymbol{\epsilon}_{y \cdot xzw}$ is an $n$-dimensional vector of error variables. Here, we assume that elements of $\boldsymbol{\epsilon}_{y \cdot xzw}$ are independent and identically distributed with mean zero and variance $\sigma_{yy \cdot xzw} < \infty$. In this paper, we also assume that a treatment variable, a response variable and explanatory variables are standardized to a sample mean of zero and a variance of one in advance. Here, we consider a situation where (i) $\boldsymbol{Z} \cup \boldsymbol{W}$ is a set of covariates satisfying the back-door criterion relative to $(X, Y)$, (ii) $\boldsymbol{Z}$ is a subset of confounders selected from prior causal knowledge (possibly an empty set, a sufficient set of confounders, or an insufficient set of confounders), and (iii) $\boldsymbol{W}$ is a set of covariates for which it is uncertain which covariate should be added to $\boldsymbol{Z}$ as a confounder, or we know that a given set of covariates satisfies the back-door criterion but the OLS method is not applicable to estimating total effects using such a set because of the multicollinearity/high-dimensional data problem.

Then, for a smaller subset of $\boldsymbol{Z} \cup \boldsymbol{W}$, if the signs of the regression coefficients of $X$ are equivalent between the regression models using $\boldsymbol{Z} \cup \boldsymbol{W}$ and a selected smaller set, the regression model using such a subset will not provide misleading qualitative results regarding the total effects. Under the above setting, the aim of this paper is to derive a consistent or less-biased estimator of the total effect.

This paper mainly focuses on a situation where the sum of squares matrix of $\{X\} \cup \boldsymbol{Z}$ is invertible but that of $\{X\} \cup \boldsymbol{Z} \cup \boldsymbol{W}$ is not, because if it is invertible then the total effect is estimable by the OLS method [Pearl, 2009].

## 3.2 PAL$_p$MA ESTIMATOR

We let $s_{xy}$, $S_{zw}$ and $S_{xz}$ be the sum of cross-products between $X$ and $Y$, the sum of the cross-product matrix between $\boldsymbol{Z}$ and $\boldsymbol{W}$ and the sum of the cross-product vectors between $X$ and $\boldsymbol{Z}$, respectively. In addition, we let $s_{xx}$, $S_{zz}$ and $I_{q,q}$ be the sum of squares of $X$, the sum of squares matrix of $\boldsymbol{Z}$ and a $q \times q$ identity matrix, respectively. Furthermore, $s_{xx \cdot zw}$, $S_{xw \cdot z}$ and $S_{ww \cdot z}$ are the conditional sum of squares of $X$ given $\boldsymbol{Z}$ and $\boldsymbol{W}$, the conditional sum of the cross-product vector between $X$ and $\boldsymbol{W}$ given $\boldsymbol{Z}$ and the conditional sum of squares matrix of $\boldsymbol{W}$ given $\boldsymbol{Z}$, respectively. Similar notation is used for the remaining sum of squares/cross-products. Then, the proposed method, PAL$_p$MA, is formulated as follows:

Let

$$\begin{pmatrix} \hat{\beta}_{yx \cdot zw} \\ \hat{B}_{yz \cdot xw} \\ \hat{B}_{yw \cdot xz} \end{pmatrix} = \begin{pmatrix} s_{xx} & S_{xz} & S_{xw} \\ S_{xz}^T & S_{zz} & S_{zw} \\ S_{xw}^T & S_{zw}^T & S_{ww} \end{pmatrix}^{-1} \begin{pmatrix} s_{xy} \\ S_{zy} \\ S_{wy} \end{pmatrix} \quad (3)$$

when the sum of squares matrix of the explanatory variables is invertible, and

$$\begin{pmatrix} \tilde{\beta}_{yx \cdot zw} \\ \tilde{B}_{yz \cdot xw} \\ \tilde{B}_{yw \cdot xz} \end{pmatrix} = \begin{pmatrix} s_{xx} & S_{xz} & S_{xw} \\ S_{xz}^T & S_{zz} & S_{zw} \\ S_{xw}^T & S_{zw}^T & \lambda I_{q,q} + S_{ww} \end{pmatrix}^{-1} \begin{pmatrix} s_{xy} \\ S_{zy} \\ S_{wy} \end{pmatrix} \quad (4)$$

for $\lambda > 0$ when the sum of squares matrix of the explanatory variables is not invertible. Then, for $p = 1, 2$, consider the loss function

$$L_p(\beta_{yx \cdot zw}, B_{yz \cdot xw}, B_{yw \cdot xz})$$
$$= \frac{1}{2}||\boldsymbol{y} - \boldsymbol{x}\beta_{yx \cdot zw} - \boldsymbol{z}B_{yz \cdot xw} - \boldsymbol{w}B_{yw \cdot xz}||_2^2$$
$$+ \lambda_p||\boldsymbol{\gamma} \odot B_{yw \cdot xz}||_p^p, \quad (5)$$

where $\boldsymbol{\gamma} = (\gamma_1, \gamma_2, ..., \gamma_q)^T$ is a weight vector such that

$$\boldsymbol{\gamma} = \left( \frac{1}{|\tilde{\beta}_{yw_1 \cdot xzw_{(1)}}|^\xi}, \cdots, \frac{1}{|\tilde{\beta}_{yw_q \cdot xzw_{(q)}}|^\xi} \right)^T \quad (6)$$

for the non-invertible sum of squares matrix of the explanatory variables with tuning parameter $\xi \geq 0$, and

$$\boldsymbol{\gamma} = \left( \frac{1}{|\hat{\beta}_{yw_1 \cdot xzw_{(1)}}|^\xi}, \cdots, \frac{1}{|\hat{\beta}_{yw_q \cdot xzw_{(q)}}|^\xi} \right)^T \quad (7)$$

for the invertible sum of squares matrix of the explanatory variables with tuning parameter $\xi \geq 0$, where the superscript "$T$" stands for a transposed vector/matrix. In addition, $\odot$ refers to the Hadamard product. $|| \cdot ||_p$ denotes the $L_p$ norm, and $\lambda_p$ is called a regularization parameter corresponding to the $L_p$ norm ($\lambda_p \geq 0$). $| \cdot |$ stands for the absolute value. The loss function (equation (5)) is different from standard $L_p$-regularized loss functions in the sense that the regularization parameter $\lambda_p$ is not assigned to $\beta_{yx \cdot zw}$ or $B_{yz \cdot xw}$. In this sense, equation (5) is called a partially adaptive $L_p$-regularized loss function in this paper. Here, under the assumption the sum of squares matrix of explanatory variables $\{X\} \cup \boldsymbol{Z} \cup \boldsymbol{W}$ is invertible, letting $\lambda_p = 0$, $\beta_{yx \cdot zw}$, $B_{yz \cdot xw}$ and $B_{yw \cdot xz}$ that minimize equation (5) yield equation (3), i.e., the OLS estimators $\hat{\beta}_{yx \cdot zw}$, $\hat{B}_{yz \cdot xw}$ and $\hat{B}_{yw \cdot xz}$ of equation (2), respectively. Letting $\lambda_2 = \lambda$ and $\xi = 0$, $\beta_{yx \cdot zw}$, $B_{yz \cdot xw}$ and $B_{yw \cdot xz}$ that minimize equation (5) yield equation (4), i.e., the ridge-type estimators $\tilde{\beta}_{yx \cdot zw}$, $\tilde{B}_{yz \cdot xw}$ and $\tilde{B}_{yw \cdot xz}$ of equation (2), respectively.

For $p = 1$ and $\lambda_1 > 0$, $\beta_{yx \cdot zw}$, $B_{yz \cdot xw}$ and $B_{yw \cdot xz}$ that minimize equation (5) are called PAL$_1$MA estimators, denoted by $\check{\beta}_{yx \cdot zw}^\dagger$, $\check{B}_{yz \cdot xw}^\dagger$ and $\check{B}_{yw \cdot xz}^\dagger$, respectively. If $\boldsymbol{W}$ is an active set for given $\lambda_1 > 0$, that is, a subset of explanatory variables with nonzero regression coefficients, but does not include any elements of $\{X\} \cup \boldsymbol{Z}$ (i.e., any $i$-th element of $\check{B}_{yw \cdot xz}^\dagger$ does not take the value zero for given $\lambda_1 > 0$ ), and letting $q$ be the number of explanatory variables in the active set $\boldsymbol{W}$, then under the assumption that the sum of squares matrix of explanatory variables $\{X\} \cup \boldsymbol{Z} \cup \boldsymbol{W}$ is invertible, $\check{\beta}_{yx \cdot zw}^\dagger$ is given by

$$\check{\beta}_{yx \cdot zw}^\dagger = \hat{\beta}_{yx \cdot zw} + \frac{\lambda_1}{s_{xx \cdot zw}} \hat{B}_{xw \cdot z}^T \boldsymbol{\gamma} \odot \text{sign}(\check{B}_{yw \cdot xz}^\dagger). \quad (8)$$

Here, $\hat{B}_{xw \cdot z}$ is given by $\hat{B}_{xw \cdot z} = S_{ww \cdot z}^{-1} S_{wx \cdot z}$. In addition, for a $q$-dimensional vector $\boldsymbol{a} = (a_1, a_2, ..., a_q)^T$, $\text{sign}(\boldsymbol{a}) = (\text{sign}(a_1), \text{sign}(a_2), ..., \text{sign}(a_q))^T$, where

$$\text{sign}(a_i) = \begin{cases} 1 & : a_i > 0 \\ 0 & : a_i = 0 \\ -1 & : a_i < 0 \end{cases} \quad (9)$$

for $i = 1, 2, ..., q$. For $p = 2$ and $\lambda_2 > 0$, $\beta_{yx \cdot zw}$, $B_{yz \cdot xw}$ and $B_{yw \cdot xz}$ that minimize equation (5) are called PAL$_2$MA estimators, denoted by $\tilde{\beta}_{yx \cdot zw}^\dagger$, $\tilde{B}_{yz \cdot xw}^\dagger$

and $\tilde{B}_{yw \cdot xz}^{\dagger}$, respectively. Then, $\tilde{\beta}_{yx \cdot zw}^{\dagger}$ is given by

$$\tilde{\beta}_{yx \cdot zw}^{\dagger} = \hat{\beta}_{yx \cdot zw}$$
$$+ \frac{\lambda_2 \hat{B}_{yw \cdot xz}^{T} (S_{ww \cdot z} + \lambda_2 \mathrm{diag}(\boldsymbol{\gamma}))^{-1} S_{wx \cdot z}}{s_{xx \cdot z} - S_{xw \cdot z} (S_{ww \cdot z} + \lambda_2 \mathrm{diag}(\boldsymbol{\gamma}))^{-1} S_{wx \cdot z}}, \quad (10)$$

where $\mathrm{diag}(\boldsymbol{\gamma})$ is a diagonal matrix whose $(i, i)$ element corresponds to the $i$-th element of $\boldsymbol{\gamma}$ ($i = 1, 2, ..., q$).

### 3.3  $\mathbf{L}_p$ COLLAPSIBILITY

In this section, we extend the concept of collapsibility from the framework of traditional regression analysis to regularized regression analysis as follows:

**Definition 3** (*$L_p$ Collapsibility*) *For a given $p$, $\boldsymbol{W}$ is said to be $L_p$ collapsible with the regression coefficient of $X$ on $Y$ in regression model (2) when the coefficient does not depend on $\boldsymbol{W}$ or the regularization parameter $\lambda_p$. In particular, when $\boldsymbol{W}$ is $L_p$ collapsible with the regression coefficient of $X$ on $Y$ in regression model (2) for $p = 1, 2$, $\boldsymbol{W}$ is said to be collapsible with the regression coefficient of $X$ on $Y$ in regression model (2).*

From equations (8) and (10), the following theorem is derived immediately:

**Theorem 1** *For $p = 1, 2$, when the sum of squares matrix of $\boldsymbol{Z} \cup \boldsymbol{W}$ is invertible, if $S_{xw \cdot z} = \mathbf{0}_q$ holds, $\boldsymbol{W}$ is collapsible with the regression coefficient of $X$ on $Y$ in regression model (2), i.e., we have*

$$\check{\beta}_{yx \cdot zw}^{\dagger} = \tilde{\beta}_{yx \cdot zw}^{\dagger} = \hat{\beta}_{yx \cdot zw} = \hat{\beta}_{yx \cdot z}. \quad (11)$$

*Particularly, if $X$ is conditionally independent of $\boldsymbol{W}$ given $\boldsymbol{Z}$, we have*

$$E(\check{\beta}_{yx \cdot zw}^{\dagger}) = E(\tilde{\beta}_{yx \cdot zw}^{\dagger}) = E(\hat{\beta}_{yx \cdot zw}) = E(\hat{\beta}_{yx \cdot z}). \quad (12)$$

Note that $\boldsymbol{W}$ is assumed to be an active set for $p = 1$ in Theorem 1.

**Theorem 2** *For $p = 2$, when the sum of squares matrix of $\{X\} \cup \boldsymbol{Z} \cup \boldsymbol{W}$ is invertible, if $S_{yw \cdot xz} = \mathbf{0}_q$ holds, $\boldsymbol{W}$ is $L_2$ collapsible with the regression coefficient of $X$ on $Y$ in regression model (2), i.e., we have*

$$\tilde{\beta}_{yx \cdot zw}^{\dagger} = \hat{\beta}_{yx \cdot zw} = \hat{\beta}_{yx \cdot z}. \quad (13)$$

*Particularly, if $Y$ is conditionally independent of $\boldsymbol{W}$ given $X$ and $\boldsymbol{Z}$, we have*

$$E(\tilde{\beta}_{yx \cdot zw}^{\dagger}) = E(\hat{\beta}_{yx \cdot zw}) = E(\hat{\beta}_{yx \cdot z}). \quad (14)$$

Generally, standard regularized regression analysis does not provide consistent estimators of the regression coefficients. In contrast, from Theorem 1, for $p = 1, 2$, PAL$_p$MA provides the consistent estimator of the regression coefficient of $X$ on $Y$ if $X$ and $\boldsymbol{W}$ are conditionally independent given $\boldsymbol{Z}$, regardless of the regularization parameter. In other words, when $\boldsymbol{W}$ is $L_p$ collapsible with the regression coefficient of $X$ on $Y$ and $\boldsymbol{Z}$ satisfies the back-door criterion relative to $(X, Y)$ in regression model (2), PAL$_p$MA can provide a consistent estimator of the total effect. On the other hand, when $X$ is not conditionally independent of $\boldsymbol{W}$ given $\boldsymbol{Z}$, PAL$_p$MA may provide a biased estimator of the regression coefficient of $X$ on $Y$.

To reduce the bias, consider a partially adaptive $L_2$-regularized loss function with a weight vector $\boldsymbol{\gamma}^*$ and a tuning parameter $\xi^*$ such that $\boldsymbol{x}$ and $\boldsymbol{y}$ are replaced by an empty set and $\boldsymbol{x}$ in equation (5), respectively. Letting $\tilde{B}_{xw \cdot z}^{\dagger}$ and $\tilde{B}_{xz \cdot w}^{\dagger}$ be PAL$_2$MA estimators of $B_{xw \cdot z}$ and $B_{xz \cdot w}$ derived from such a loss function, respectively, from equation (8), we formulate the modified PAL$_1$MA estimator of $\beta_{yx \cdot zw}$ as

$$\check{\beta}_{yx \cdot zw}^{*} = \check{\beta}_{yx \cdot zw}^{\dagger} - \frac{\lambda_1}{\tilde{s}_{xx \cdot zw}^{\dagger}} \tilde{B}_{xw \cdot z}^{\dagger T} \boldsymbol{\gamma} \odot \mathrm{sign}(\check{B}_{yw \cdot xz}^{\dagger}), \quad (15)$$

$$\tilde{s}_{xx \cdot zw}^{\dagger} = ||\boldsymbol{x} - \boldsymbol{z} \tilde{B}_{xz \cdot w}^{\dagger} - \boldsymbol{w} \tilde{B}_{xw \cdot z}^{\dagger}||_2^2 \quad (16)$$

for an active set $\boldsymbol{W}$. When the sum of squares matrix of $\{X\} \cup \boldsymbol{Z} \cup \boldsymbol{W}$ is invertible, we have

$$\check{\beta}_{yx \cdot zw}^{*} = \check{\beta}_{yx \cdot zw}^{\dagger} - \frac{\lambda_1}{\tilde{s}_{xx \cdot zw}^{\dagger}} \tilde{B}_{xw \cdot z}^{\dagger T} \boldsymbol{\gamma} \odot \mathrm{sign}(\check{B}_{yw \cdot xz}^{\dagger})$$

$$= \hat{\beta}_{yx \cdot zw} + \lambda_1 \left( \frac{1}{s_{xx \cdot zw}} \hat{B}_{xw \cdot z} - \frac{1}{\tilde{s}_{xx \cdot zw}^{\dagger}} \tilde{B}_{xw \cdot z}^{\dagger} \right)^{T}$$

$$\times \boldsymbol{\gamma} \odot \mathrm{sign}(\check{B}_{yw \cdot xz}^{\dagger}). \quad (17)$$

Thus, when $\boldsymbol{Z} \cup \boldsymbol{W}$ satisfies the back-door criterion, if $\hat{B}_{xw \cdot z} = \tilde{B}_{xw \cdot z}^{\dagger}$ and $s_{xx \cdot zw} = \tilde{s}_{xx \cdot zw}^{\dagger}$ hold (i.e., these estimators are not dependent on the regularization parameter), then the total effect is estimated by $\check{\beta}_{yx \cdot zw}^{*}$. In addition, since we have

$$\check{\beta}_{yx \cdot zw}^{*} = \hat{\beta}_{yx \cdot zw} + \lambda_1 \hat{B}_{xw \cdot z}^{T}$$

$$\times \left( \frac{I_{q,q}}{s_{xx \cdot zw}} - \frac{S_{ww \cdot z} (S_{ww \cdot z} + \lambda_2 \mathrm{diag}(\boldsymbol{\gamma}^*))^{-1}}{\tilde{s}_{xx \cdot zw}^{\dagger}} \right)$$

$$\times \boldsymbol{\gamma} \odot \mathrm{sign}(\check{B}_{yw \cdot xz}), \quad (18)$$

if $S_{xw \cdot z} = \mathbf{0}_q$, the total effect is also estimated by $\check{\beta}_{yx \cdot zw}^{*}$.

On the contrary, even when the sum of squares matrix of $\{X\} \cup \boldsymbol{Z} \cup \boldsymbol{W}$ is not invertible, by taking a small value of $\lambda_2 > 0$ such that $S_{ww \cdot z} + \lambda_2 \mathrm{diag}(\boldsymbol{\gamma}^*)$ is invertible in equation (18), the modified PAL$_1$MA can provide the less-biased estimator of the total effects.

Hereafter, the modified PAL$_1$MA estimator is merely called the PAL$_1$MA estimator.

### 3.4 I-PROGLES

Similar to standard regularized regression analysis such as LASSO, adaptive LASSO and elastic net, it is difficult to provide the explicit formula of the PAL$_1$MA estimator of the regression coefficient of $X$ on $Y$, since equation (5) includes the non-differentiable term $||\boldsymbol{\gamma} \odot B_{yw \cdot xz}||_1^1$; the optimization algorithm is needed to derive the PAL$_1$MA estimator. Here, note that standard LASSO algorithms such as least angle regression [Efron et al, 2004] and generalized path seeking [Friedman, 2012] are not applicable to achieve our aim since neither $\beta_{yx \cdot zw}$ nor $B_{yz \cdot xw}$ are regularized in equation (5).

To derive the PAL$_1$MA estimator $\check{\beta}_{yx \cdot zw}^*$, we propose a novel optimization algorithm that adopts the idea of the block coordinate relaxation method [Sardy et al, 2000]: "integrated algorithm of PROximal Gradient method and LEast Squares method" (i-PROGLES). i-PROGLES, which is shown in Algorithm 1, can be considered the integrated iterative algorithm of the proximal gradient method [Daubechies et al, 2004] and the OLS method. i-PROGLES enables us to include both the treatment variable and some of important confounders in the regression model, regardless of the value taken by the regularization parameters. In addition, if we know that a set of covariates satisfies the back-door criterion, the solution path with such information can be utilized as the criteria of parameter tuning of i-PROGLES to include the set of covariates.

To construct i-PROGLES, let $\boldsymbol{w}_i$ be an $n$-dimensional observation vector of the $i$-th explanatory variable $W_i$ of $\boldsymbol{W}$ ($W_i \in \boldsymbol{W} : i = 1, 2, ..., q$). In addition, based on the weight vector $\boldsymbol{\gamma}$ from equations (6) and (7), we define the $n \times q$ matrix $\boldsymbol{w}^\sharp$ and $B_{yw \cdot xz}^\sharp$ as $\boldsymbol{w}^\sharp = \left( \gamma_1^{-1} \boldsymbol{w}_1, \gamma_2^{-1} \boldsymbol{w}_2, \ldots, \gamma_q^{-1} \boldsymbol{w}_q \right)$ and $\boldsymbol{\gamma} \odot B_{yw \cdot xz}$, respectively. Then, for $p = 1$, equation (5) is reformulated as

$$L_1^\sharp(\beta_{yx \cdot zw}, B_{yz \cdot xw}, B_{yw \cdot xz}^\sharp)$$
$$= \frac{1}{2}||\boldsymbol{y} - \boldsymbol{x}\beta_{yx \cdot zw} - \boldsymbol{z}B_{yz \cdot xw} - \boldsymbol{w}^\sharp B_{yw \cdot xz}^\sharp||_2^2$$
$$+ \lambda_1 ||B_{yw \cdot xz}^\sharp||_1^1. \quad (19)$$

Here, $B_{yw \cdot xz}^\sharp[0]$ is defined as the solution of equation (19) given $\beta_{yx \cdot zw} = \hat{\beta}_{yx \cdot z}(= \beta_{yx \cdot zw}[0])$ and $B_{yz \cdot xw} = \hat{B}_{yz \cdot x}(= B_{yz \cdot xw}[0])$. Based on equation (19), in the first substep of the $k+1$-th step ($k \geq 0$), we evaluate $B_{yw \cdot xz}^\sharp$ as the solution of the naive LASSO given $\beta_{yx \cdot zw} = \beta_{yx \cdot zw}[k]$ and $B_{yz \cdot xw} = B_{yz \cdot xw}[k]$:

$$B_{yw \cdot xz}^\sharp[k + 1]$$

$$= \underset{B}{\mathrm{argmin}} \left( L_1^\sharp(\beta_{yx \cdot zw}[k], B_{yz \cdot xw}[k], B) \right). \quad (20)$$

Here, letting $S_{ww}^\sharp$, $S_{yw}^\sharp$, $S_{wx}^\sharp$ and $S_{wz}^\sharp$ be the sum of squares matrix of $\boldsymbol{W}^\sharp$, the sum of cross-products vector between $Y$ and $\boldsymbol{W}^\sharp$, the sum of cross-products vector between $\boldsymbol{W}^\sharp$ and $X$ and the sum of cross-products matrix between $\boldsymbol{W}^\sharp$ and $\boldsymbol{Z}$, respectively, and

$$f^\sharp(\beta_{yx \cdot zw}, B_{yz \cdot xw}, B_{yw \cdot xz})$$
$$= \frac{1}{2}||\boldsymbol{y} - \boldsymbol{x}\beta_{yx \cdot zw} - \boldsymbol{z}B_{yz \cdot xw} - \boldsymbol{w}^\sharp B_{yw \cdot xz}||_2^2, \quad (21)$$

$B_{yw \cdot xz}^\sharp[k + 1]$ is formulated by

$$B_{yw \cdot xz}^\sharp[k + 1] = \mathrm{prox}_{\eta\lambda_1} \left( B_{yw \cdot xz}^\sharp[k] - \eta \quad (22) \right.$$
$$\left. \times \frac{\partial}{\partial B} f^\sharp \left( \beta_{yx \cdot zw}[k], B_{yz \cdot xw}[k], B \right)_{B = B_{yw \cdot xz}^\sharp[k]} \right),$$

which is straightforward from equation (20) through the proximal gradient method [Daubechies et al, 2004] given $\beta_{yx \cdot zw}[k]$ and $B_{yz \cdot xw}[k]$. In this paper, $\mathrm{prox}_a(b)$ is defined as

$$\mathrm{prox}_a(b) = \begin{cases} b - a & : b \geq a \\ 0 & : -a < b < a \, . \\ b + a & : b \leq -a \end{cases} \quad (23)$$

In addition, noting

$$\frac{\partial}{\partial B} f^\sharp \left( \beta_{yx \cdot zw}[k], B_{yz \cdot xw}[k], B \right)_{B = B_{yw \cdot xz}^\sharp[k]} \quad (24)$$
$$= S_{wx}^\sharp \beta_{yx \cdot zw}[k] + S_{wz}^\sharp B_{yz \cdot xw}[k] + S_{ww}^\sharp B_{yw \cdot xz}^\sharp[k] - S_{wy}^\sharp,$$

we have

$$B_{yw \cdot xz}^\sharp[k + 1] = \mathrm{prox}_{\eta\lambda_1}(B_{yw \cdot xz}^\sharp[k] - \eta \ (S_{wx}^\sharp \beta_{yx \cdot zw}[k]$$
$$+ S_{wz}^\sharp B_{yz \cdot xw}[k] + S_{ww}^\sharp B_{yw \cdot xz}^\sharp[k] - S_{wy}^\sharp)), \quad (25)$$

where $\eta$ satisfies $\eta \leq (\lambda_{\max}(S_{ww}^\sharp))^{-1}$. Here, $\lambda_{\max}(S_{ww}^\sharp)$, which is the maximum eigenvalue of $S_{ww}^\sharp$, corresponds to the Lipschitz constant with respect to $(\partial/\partial B_{yw \cdot xz})f^\sharp$.

In the second substep of the $k+1$-th step, we evaluate $\beta_{yx \cdot zw}[k + 1]$ and $B_{yz \cdot xw}[k + 1]$ by the OLS method given $B_{yw \cdot xz} = B_{yw \cdot xz}[k + 1]$:

$$\left( \beta_{yx \cdot zw}[k + 1], B_{yz \cdot xw}^T[k + 1] \right)^T$$
$$= \underset{b,B}{\mathrm{argmin}} \left( f^\sharp(b, B, B_{yw \cdot xz}[k + 1]) \right)$$
$$\quad (26)$$
$$= \begin{pmatrix} s_{xx} & S_{xz} \\ S_{xz}^T & S_{zz} \end{pmatrix}^{-1} \begin{pmatrix} \boldsymbol{x}^T \\ \boldsymbol{z}^T \end{pmatrix} (\boldsymbol{y} - \boldsymbol{w}B_{yw \cdot xz}[k + 1]) \, .$$

Regarding the convergence of i-PROGLES, the following theorem can be derived:

---

**Algorithm 1** : i-PROGLES  (both $\lambda_2$ and $\xi_2$ are used to derive $\tilde{B}^{\dagger}_{xw \cdot z}$ and $\tilde{s}^{\dagger}_{xx \cdot zw}$)

**Input:** $\boldsymbol{x}$, $\boldsymbol{y}$, $\boldsymbol{z}$ and $\boldsymbol{w}$, $k^* > 0$, $\lambda_1 \geq 0$, $\lambda_2 \geq 0$, $\xi_1 > 0$, $\xi_2 > 0$

$$\beta_{yx \cdot zw}[0] = \hat{\beta}_{yx \cdot z}, \;\; B_{yz \cdot xw}[0] = \hat{B}_{yz \cdot x}$$

$$B^{\sharp}_{yw \cdot xz}[0] = \underset{B}{\operatorname{argmin}} \left( \frac{1}{2} ||\boldsymbol{y} - \boldsymbol{x}\hat{\beta}_{yx \cdot z} - \boldsymbol{z}\hat{B}_{yz \cdot x} - \boldsymbol{w}^{\sharp}B||^2_2 + \lambda_1 ||B||_1 \right)$$

Calculate the weight vector: If the sum of squares matrix of the explanatory variables is not invertible, set

$$\boldsymbol{\gamma} = \left( \frac{1}{|\tilde{\beta}_{yw_1 \cdot xzw_{(1)}}|^{\xi_1}}, \frac{1}{|\tilde{\beta}_{yw_2 \cdot xzw_{(2)}}|^{\xi_1}}, \cdots, \frac{1}{|\tilde{\beta}_{yw_q \cdot xzw_{(q)}}|^{\xi_1}} \right)^T$$

If the sum of squares matrix of the explanatory variables is invertible, set

$$\boldsymbol{\gamma} = \left( \frac{1}{|\hat{\beta}_{yw_1 \cdot xzw_{(1)}}|^{\xi_1}}, \frac{1}{|\hat{\beta}_{yw_2 \cdot xzw_{(2)}}|^{\xi_1}}, \cdots, \frac{1}{|\hat{\beta}_{yw_q \cdot xzw_{(q)}}|^{\xi_1}} \right)^T$$

1: **for** $k = 0$ to $k^*$ **do**
2:    Set
$$\eta \leq (\lambda_{\max}(S^{\sharp}_{ww}))^{-1}$$
$$B^{\sharp}_{yw \cdot xz}[k+1] = \operatorname{prox}_{\eta \lambda_1}(B^{\sharp}_{yw \cdot xz}[k] - \eta(S^{\sharp}_{wx}\beta_{yx \cdot zw}[k] + S^{\sharp}_{wz}B_{yz \cdot xw}[k] + S^{\sharp}_{ww}B^{\sharp}_{yw \cdot xz}[k] - S^{\sharp}_{wy}))$$

3:    Set
$$B_{yw \cdot xz}[k+1] = \left( \gamma_1^{-1}\beta^{\sharp}_{yw_1 \cdot xzw_{(1)}}[k+1], \gamma_2^{-1}\beta^{\sharp}_{yw_2 \cdot xzw_{(2)}}[k+1] \ldots, \gamma_q^{-1}\beta^{\sharp}_{yw_q \cdot xzw_{(q)}}[k+1] \right)^T$$

4:    Set
$$\beta_{yx \cdot zw}[k+1] = \hat{\beta}_{yx \cdot z} - \hat{B}_{wx \cdot z}B_{yw \cdot xz}[k+1], \quad B_{yz \cdot xw}[k+1] = \hat{B}_{yz \cdot x} - \hat{B}_{wz \cdot x}B_{yw \cdot xz}[k+1]$$

5: **end for**
6: Set
$$\breve{\beta}^*_{yx \cdot zw} = \beta_{yx \cdot zw}[k^*+1] - \frac{\lambda_1}{\tilde{s}^{\dagger}_{xx \cdot zw}} \tilde{B}^{\dagger T}_{xw \cdot z}\boldsymbol{\gamma} \odot \operatorname{sign}(B_{yw \cdot xz}[k^*+1])$$

7: **return** $\breve{\beta}^*_{yx \cdot zw}$

---

**Theorem 3** *Let $\{\beta_{yx \cdot zw}[k]\}_{k \geq 0}$, $\{B_{yz \cdot xw}[k]\}_{k \geq 0}$ and $\{B_{yw \cdot xz}[k]\}_{k \geq 0}$ be the sequences of $\beta_{yx \cdot zw}$, $B_{yz \cdot xw}$ and $B_{yw \cdot xz}$, respectively, generated by i-PROGLES, and let $\boldsymbol{u} = (\boldsymbol{x}, \boldsymbol{z})$. When $\beta^*_{yx \cdot zw}$, $B^*_{yz \cdot xw}$ and $B^*_{yw \cdot xz}$ minimize equation (19) regarding $\beta_{yx \cdot zw}$, $B_{yz \cdot xw}$ and $B_{yw \cdot xz}$, respectively, there exists the natural number $K$ for any $\epsilon > 0$ such that*

$$L_1\left( \beta^*_{yx \cdot zw}, B^*_{yz \cdot xw}, B^*_{yw \cdot xz} \right)$$
$$- L_1\left( \beta_{yx \cdot zw}[k+1], B_{yz \cdot xw}[k+1], B_{yw \cdot xz}[k+1] \right)$$
$$\leq \frac{\lambda max(S^{\sharp}_{ww})}{2k} ||B^{\sharp}_{yw \cdot xz}[0] - B^{\sharp *}_{yw \cdot xz}||^2_2$$

$$+ \frac{\lambda max(S_{uu})}{2} \lambda max(S^{\sharp}_{wu}S^{-2}_{uu}S^{\sharp}_{uw})\epsilon. \tag{27}$$

*holds for any $k \geq K$, where $B^{\sharp}_{yw \cdot xz}[k] = \boldsymbol{\gamma} \odot B_{yw \cdot xz}[k]$ and $B^{\sharp *}_{yw \cdot xz} = \boldsymbol{\gamma} \odot B^*_{yw \cdot xz}$.*

The proof is given in the Supplementary Material.

## 4   NUMERICAL EXPERIMENT

In this section, we present a numerical experiment to compare the performance of LASSO, adaptive LASSO,

Table 1. Results based on cross-validation.

| | (a) $\tau_{yx} = 0.474$ | | | | | parameter settings | | | | |
| | mean | bias | mse | sd | sign | $\lambda$ | $\xi$ | $\phi$ | $\lambda_1$ | $\xi_1$ |
|---|---|---|---|---|---|---|---|---|---|---|
| LASSO | 0.1812 | 0.2929 | 0.1012 | 0.1238 | 0.8824 | 0.0830 | - | - | - | - |
| adaptive LASSO | 0.2736 | 0.2006 | 0.0776 | 0.1934 | 0.8932 | 3.4300 | 1.7000 | - | - | - |
| Elastic net | 0.2101 | 0.2641 | 0.0807 | 0.1047 | 0.9664 | 0.0780 | - | 0.5500 | - | - |
| MCP | 0.2290 | 0.2451 | 0.0862 | 0.1617 | 0.8462 | 0.0600 | 19.5000 | - | - | - |
| SCAD | 0.1909 | 0.2832 | 0.1032 | 0.1517 | 0.8216 | 0.0860 | 15.5000 | - | - | - |
| PAL$_1$MA | 0.4486 | 0.0256 | 0.0640 | 0.2516 | 0.9746 | - | - | - | 0.0100 | 0.1000 |
| OLS | 0.4717 | 0.0025 | 0.2961 | 0.5441 | 0.8154 | - | - | - | - | - |

mean: sample mean; bias: bias between the true value and the sample mean; mse: mean squared error; sd: standard deviation; sign: coincidence rate between the signs of the true value and the estimates; $\lambda$, $\lambda_1$: regularization parameters; $\xi$, $\xi_1$: tuning parameters; $\phi$: mixing parameter. The regularization parameter $\lambda_2$ and tuning parameter $\xi_2$ are selected as $\lambda_2 = 0.0014$, $\xi_2 = 0.0013$. Refer to the Supplementary Material for the selection of these parameters.

elastic net, SCAD, MCP, OLS and PAL$_1$MA. For simplicity, letting $X$ and $Y$ be the treatment variable and the response variable, respectively, consider the linear SCM with 42 explanatory variables for $Y$ in the form of

$$\left. \begin{array}{l} Y = \alpha_{yx}X + \alpha_{yz_1}Z_1 + \alpha_{yz_2}Z_2 + A_{yw}\boldsymbol{W} + \epsilon_y \\ X = \alpha_{xz_1}Z_1 + \alpha_{xz_2}Z_2 + \epsilon_x \end{array} \right\} \quad (28)$$

for Fig. 1($\boldsymbol{W}$ includes 39 variables). In this setting, we assume that $\{Z_1, Z_2\}$ satisfies the back-door criterion relative to $(X, Y)$ and the path coefficients of $\{Z_2\} \cup \boldsymbol{W}$ on $Y$ are regularized but $Z_1$ is not. Then, Theorem 1 does not hold, and the estimated total effect may be biased.

To set up a simulation, we first construct the population variance-covariance matrix. To eliminate the arbitrariness, the true values of the path coefficients $\alpha_{yx}, \alpha_{yz_1}, \alpha_{yz_2}, A_{yw} = (\alpha_{yw_1}, ..., \alpha_{yw_{39}}), \alpha_{xz_1}$ and $\alpha_{xz_2}$ are randomly and independently determined according to the uniform distribution with the interval $[-3, 3]$. In addition, we assume that (i) the random disturbances $\epsilon_x$ and $\epsilon_y$ independently follow the normal distribution with mean zero and variance one, and (ii) the random disturbances are also independent of their nondescendants. Furthermore, the population variance-covariance matrices of $\{Z_1, Z_2\} \cup \boldsymbol{W}$ are randomly determined according to Pourahmadi and Wang [2015].

We generated 30 random samples of 42 variables from a multivariate normal distribution with a zero mean vector and the above variance-covariance matrix for 5000 replications. Table 1 shows the basic statistics of the total effects estimated by LASSO, adaptive LASSO, elastic net, SCAD, MCP, OLS and PAL$_1$MA

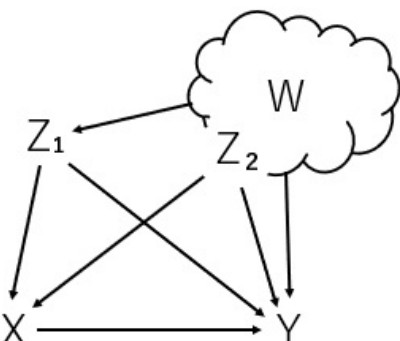

Fig. 1. Causal diagram

based on the given sample size of 30 for each parameter setting. Regarding the parameter tuning for regularized regression analysis, see the Supplementary Material. Here, for the OLS method, we select a set of covariates based on prior causal knowledge; i.e., $\{Z_1, Z_2\}$ are selected.

From Table 1, both the PAL$_1$MA estimators and the OLS estimators are almost consistent with the true values of the total effects, but the other regularized regression methods yield highly biased estimators. In addition, the coincidence rates between the signs of the estimated total effects and the true total effects for PAL$_1$MA are better than those for the other regression methods. From Fig. 2, the interquartile ranges of both PAL$_1$MA and OLS include the true value of the total effects, but the other regularized regression analyses do not include this value of the total effects. For further discussion on the simulation experiments, see the

Supplementary Material.

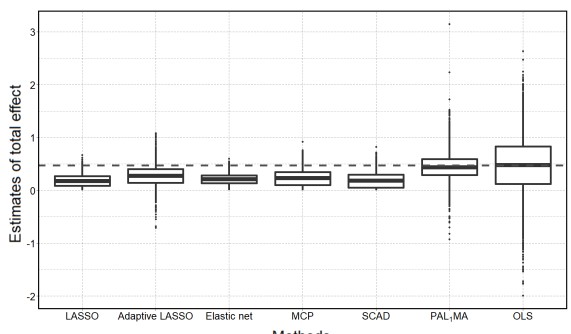

Fig. 2. Boxplots of the estimated total effects. The dashed lines show the true total effects.

## 5 CONCLUSION

In current situations where advanced artificial intelligence (AI) technology enables us to collect large datasets, it would not be so difficult to observe a large number of covariates. In such situations, it would be reasonable to consider that such a set of covariates satisfies the back-door criterion to estimate the total effects. However, when multicollinearity/high-dimensional data problems occur in even this situation, it is difficult to evaluate the linear causal effects reliably. To solve this problem, we established $PAL_pMA$ to provide a consistent or less-biased estimator of the total effects. In addition, through numerical experiments and a case study in Supplementary Material, we confirmed that $PAL_1MA$ is superior to other estimation methods. The results of this paper are applicable to evaluating the direct effect in the framework of regression models through the "single-door criterion" [Pearl, 2009]. The results of this paper would also help us to obtain the reliable evaluation of the mean of the response variable when conducting the external intervention (e.g., Kuroki and Nanmo 2020, Nanmo and Kuroki 2021) from multicollinearity/high-dimensional data.

Finally, although $PAL_pMA$ is formulated based on linear regression models, it would be interesting to extend our approach to a wide variety of statistical models, including generalized linear models, generalized estimating equations and proportional hazards models. Such an extension would be straightforward — the objective function would be replaced with a more general form. This extension will be left for future work.

### Acknowledgement

This research was financially supported by Grant-in-Aid for Scientific Research (B) Grant Number 21H03504 and Scientific Research (C) Grant Number 19K11856.

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
