# OpenReview forum: "Partially Adaptive Regularized Multiple Regression Analysis for Estimating Linear Causal Effects"
_auai.org/UAI/2022/Conference — UAI 2022 Oral_

### Official Review · Reviewer_iqij · 2022-04-11

**Q2(1) Originality/Novelty:** 3
**Q2(2) Significance/Impact:** 4
**Q2(3) Correctness/Technical Quality:** 4
**Q2(6) Clarity Of Writing:** 4
**Q6 Overall Score:** 7
**Q8 Confidence In Your Score:** 4

**Q1 Summary And Contributions:**

The authors give a method, PALpMA, to eliminate dependence on redundant variables and give sets of variables where the backdoor criterion is satisfied, as a way of dealing with e.g. multicollinearity in high dimensional datasets in linear regression models.

**Q2 Assessment Of The Paper:**

More detailed information regarding each of these aspects is given below:

**Q2(4) Quality Of Experiments (Optional):**

4: Excellent: The experimental evaluation is comprehensive and the results are compelling.

**Q2(5) Reproducibility:**

4: Excellent: Key resources (e.g., proofs, code, data) are available and key details (e.g., proof sketches, experimental setup) are comprehensively described for competent researchers to confidently and easily reproduce the main results.

**Q3 Main Strengths:**

This is very nice. The Supplementary Materials are especially nice.

I'd say the main strength of this approach, aside from its clarity and novelty, is that it allows for parameters to be estimated with a high degree of precision, in the experiments, with small variance. The authors do not claim that this is the only way to go about selecting variables; they compare their method very favorably to a number of alternatives. Also, the theory is very clear, as are the experimental results.

**Q4 Main Weakness:**

Honestly, the scope of this paper is very nice for UAI; no particular weakness occur to me. One could further compare other methods for selecting variables, I suppose, though the methods compared here are sufficient to get the points across.

**Q5 Detailed Comments To The Authors:**

"there is a null regularization parameter is zero" is ungrammatical.

Bolding the best results per columns for Tables B and C would be very much appreciated for readability.

**Q7 Justification For Your Score:**

I don't know that this paper is ground-breaking, but it is certainly very strong and well-argued, with good experiments and results.

**Q9 Complying With Reviewing Instructions:**

1: Yes.

---

### Official Review · Reviewer_DzH1 · 2022-04-13

**Q2(1) Originality/Novelty:** 2
**Q2(2) Significance/Impact:** 2
**Q2(3) Correctness/Technical Quality:** 3
**Q2(6) Clarity Of Writing:** 3
**Q6 Overall Score:** 7
**Q8 Confidence In Your Score:** 3

**Q1 Summary And Contributions:**

The authors propose a new regularized regression method that provides "less-biased" estimates of total causal effects in linear structural equation models when dealing with multicollinearity problems.

**Q2 Assessment Of The Paper:**

More detailed information regarding each of these aspects is given below:

**Q2(4) Quality Of Experiments (Optional):**

2: Fair: The experimental evaluation is weak: important baselines are missing, or the results do not adequately support the main claims.

**Q2(5) Reproducibility:**

3: Good: Key resources (e.g., proofs, code, data) are available and key details (e.g., proofs, experimental setup) are sufficiently well-described for competent researchers to confidently reproduce the main results.

**Q3 Main Strengths:**

The authors present a good introduction, setup and motivation.

The numerical results seem promising.

**Q4 Main Weakness:**

The paper is a bit dense with lots of different notation which makes it a bit hard to follow.

**Q5 Detailed Comments To The Authors:**

Why include $\text{PAL}_2\text{MA}$ and therefore introduce even more notation if you do not provide any results for the $L_2$ case?

**Q7 Justification For Your Score:**

The authors address a relevant problem and the proposed algorithm yields promising results.

**Q9 Complying With Reviewing Instructions:**

1: Yes.

---

### Official Review · Reviewer_56W6 · 2022-04-14

**Q2(1) Originality/Novelty:** 3
**Q2(2) Significance/Impact:** 2
**Q2(3) Correctness/Technical Quality:** 3
**Q2(6) Clarity Of Writing:** 3
**Q6 Overall Score:** 7
**Q8 Confidence In Your Score:** 3

**Q1 Summary And Contributions:**

The authors propose a new regression approach to address the estimation of cause-effect relationships. They explicitly model a set of known confounders and a set of covariates (potential confounders that are highly correlated among themselves). They propose a new estimator for the total effect and characterize the conditions when it provides a consistent or less-biased estimate. They develop a new optimization algorithm and demonstrate the advantage of their approach in a numeric study.

**Q2 Assessment Of The Paper:**

More detailed information regarding each of these aspects is given below:

**Q2(4) Quality Of Experiments (Optional):**

3: Good: The experimental evaluation is adequate, and the results convincingly support the main claims.

**Q2(5) Reproducibility:**

3: Good: Key resources (e.g., proofs, code, data) are available and key details (e.g., proofs, experimental setup) are sufficiently well-described for competent researchers to confidently reproduce the main results.

**Q3 Main Strengths:**

The authors study a very relevant problem where the present regularized regression analyses don’t provide a consistent estimate for the linear cause-effect relationship in the presence of confounders.

They propose a new estimator, characterize the conditions when the new estimator provides a consistent or less-biased estimate, and develop a new optimization algorithm to derive the parameters. The work covers all the major aspects of developing a new regression analysis approach.

The numerical study is concise and direct to the point.

The paper is well organized.


**Q4 Main Weakness:**

The paper is heavy on mathematical notations and expositions. The authors can focus on the estimator with L_1 norm and only describe the results for the estimator with L_2 norm when they are different from that with L_1 norm. They can reduce the number of mathematical expositions and provide more intuitions for their approaches.

The numerical study should also provide the comparison of running times of different methods.


**Q5 Detailed Comments To The Authors:**

The authors study a very relevant problem where the present regularized regression analyses don’t provide a consistent estimate for the linear cause-effect relationship in the presence of confounders. They propose a new estimator, characterize the conditions when the new estimator provides a consistent or less-biased estimate and develop a new optimization algorithm to derive the parameters. They make theoretical contributions and provide a viable solution.
The numerical study is concise and direct to the point. But it should also provide the comparison of running times of different methods.
The paper is well organized. However, it is heavy on mathematical notations and expositions. The authors can focus on the estimator with L_1 norm and only describe the results for the estimator with L_2 norm when they are different from those for the estimator with L_1 norm. They can reduce the number of mathematical expositions and provide more intuitions for their approaches.

When the authors present their estimator in Section 3.2, the λ is subscribed according to the norm used in the loss function (Equation 5). However, in Equation 18, λ_1 and λ_2 are mixed. Is this a typo?


**Q7 Justification For Your Score:**

The authors study a very relevant problem whose solution would be useful for a subfield of AI. Their work contains both theoretical and algorithmic contributions and provides a practical solution.

The paper is well organized. The authors can improve it by reducing mathematical expositions and providing more intuitions.


**Q9 Complying With Reviewing Instructions:**

1: Yes.

---

### Decision · Program_Chairs · 2022-05-15

**Decision:**

Accept (Oral)

**Comment:**

Meta Review: This paper proposes a partially adaptive Lp-regualarized multiple regression analysis for estimating totally cause effects under a linear structural causal model. The proposed method provides a consistent or less biased estimator when a set of observed covariates satisfies the backdoor criterion. Both theoretical and empirical studies are conducted to show the effectiveness of the proposed methods.

All the reviewers agree that the proposed work is novel and makes a significant technical contribution in estimating causal effects. The new estimator is consistent or less biased, which is very nice property. The optimization algorithm is also new. There is no particular weakness raised by the reviewers, but the authors could consider refine the notations to  make it more friendly to people in other areas. I recommend acceptance of this paper given its novelty and significant technical contribution.